# Are deprivation-specific cancer survival patterns similar according to individual-based and area-based measures? A cohort study of patients diagnosed with five malignancies in England and Wales, 2008–2016

Laura M Woods ,[1] Aurélien Belot ,[1] Iain M Atherton ,[2] Lucy Elliss-Brookes ,[3] Matthew Baker,[4] Fiona C Ingleby [1]

For numbered affiliations see end of article.

**Correspondence to**
Dr Laura M Woods;
laura.woods@lshtm.ac.uk

## ABSTRACT

**Objective** To investigate if measured inequalities in cancer survival differ when using individual-based ('person') compared with area-based ('place') measures of deprivation for three socioeconomic dimensions: income, deprivation and occupation.

**Design** Cohort study.

**Setting** Data from the Office for National Statistics Longitudinal Study of England and Wales, UK, linked to the National Cancer Registration Database.

**Participants** Patients diagnosed with cancers of the colorectum, breast, prostate, bladder or with non-Hodgkin's lymphoma during the period 2008–2016.

**Primary and secondary outcome measures** Differentials in net survival between groups defined by individual wage, occupation and education compared with those obtained from corresponding area-level metrics using the English and Welsh Indices of Multiple Deprivation.

**Results** Survival was negatively associated with area-based deprivation irrespective of the type analysed, although a trend from least to most deprived was not always observed. Socioeconomic differences were present according to individually-measured socioeconomic groups although there was an absence of a consistent 'gradient' in survival. The magnitude of differentials was similar for area-based and individually-derived measures of deprivation, which was unexpected.

**Conclusion** These unique data suggest that the socioeconomic influence of 'person' is different to that of 'place' with respect to cancer outcomes. This has implications for health policy aimed at reducing inequalities. Further research could consider the separate and additional influence of area-based deprivation over individual-level characteristics (contextual effects) as well as investigate the geographic, socioeconomic and healthcare-related characteristics of areas with poor outcomes in order to inform policy intervention.

## STRENGTHS AND LIMITATIONS OF THIS STUDY

⇒ We analysed a unique, representative, cohort of England and Wales within which it was possible to classify individuals by both their area-based deprivation score and individual socioeconomic group.

⇒ We used newly derived-life tables for individual-level socioeconomic analyses to estimate underlying mortality for each individual socioeconomic group. We used a generic life table for area-based deprivation analyses because education-specific and occupation-specific life tables were unavailable, but also to prioritise the use of mortality rates derived from the Office for National Statistics Longitudinal Study itself.

⇒ We estimated individual wage on the basis of recorded occupation due to the absence of directly measured data on earnings.

⇒ Our study design enabled us to assess the relative impact of 'person' versus 'place' on socioeconomic differentials in cancer outcomes.

## INTRODUCTION

It has been widely documented that there are long-standing, persistent inequalities in cancer outcomes between individuals living in more deprived areas and those living in less deprived areas in higher-income countries.[1–7] These inequalities may partly explain why cancer survival in the UK is lower than other similar settings, where socioeconomic differentials tend to either be smaller or explicable by factors such as stage of disease at presentation.[8–10] The public health impact of these disparities is considerable as shown by the large number of associated avoidable deaths[11] and the influence this body of work has had on UK health policy over a period of more than 20 years.[12–15]

Despite their widespread use, the exact meaning of differentials observed between geographical areas with contrasting levels of deprivation have been variably interpreted.



Most often, poorer outcomes among 'persons living in deprived areas' have been treated as a proxy for poorer outcomes among 'deprived persons'[16] without consideration that area-level deprivation could have a separate and independent influence over and above an individual's own personal characteristics ('contextual effect'). This has led in turn to an implicit assumption that the inequalities observed between affluent and deprived areas are most likely diluted versions of the 'real' (unknown) differences between individuals of different socioeconomic groups, perhaps driven by the fact that, and consistent with the ecological fallacy, larger differentials are observed when the size of the geographical unit of analysis is smaller.[17 18]

Observed trends at the area-level combined with an inherent assumption of a dilution effect has thus tended to steer policy-related research into the individual domain, for example, raising individual symptom awareness in these populations,[19] increasing the probability of early stage diagnosis through screening and ensuring appropriate and effective treatment is given to patients living in more deprived localities.[19 20] Studies consistently documenting poorer outcomes in more deprived areas has also fuelled change in the way funding is allocated, causing them to be tied to specific assessments of unmet need, along with measurable, mostly individually-orientated, goals and mechanisms by which health inequalities might be narrowed.[12]

Only a relatively small body of research has examined cancer outcomes using individual, personal, measures of socioeconomic status,[21] principally because data sources suitable for such an analysis are fewer and, or, more difficult to access.[22] Socioeconomic differentials have relatively infrequently been considered as geographical phenomenon driven by locality-based factors such as travel time to hospital, access to family doctor (General Practitioner, GP) services, except in especially rural settings outside the UK.[23] Similarly, the influence of community characteristics including social capital or social cohesion has not been widely considered. These social environmental influences on health outcomes, if important, are likely to be driven by a separate set of factors to those acting purely at an individual level.

Added to this, throughout the literature on inequalities there has tended to be a singular focus on a single dimension of deprivation, normally income, or a single composite score.[22] Relatively few studies have considered whether the different dimensions of deprivation have a similar or different effect, implicitly assuming that a single measure is sufficient to examine the underlying phenomenon of interest. A broader consideration of the relative contributions of wealth, status and power[24] on cancer outcomes could help to clarify the mechanisms by which inequalities arise and are perpetuated.

Using the Office for National Statistics Longitudinal Study (ONS-LS) we have recently demonstrated that the concordance between individual socioeconomic group and the deprivation present in the small area of residence is relatively low among patients with cancer for three separate domains, but most especially for income.[25] These previous analyses suggest that interpreting area-based analyses derived from a single measure as broadly representative of individual inequalities therefore risks overlooking some important subgroups of individuals. The objective of this follow-up study was to quantify and compare inequalities according to individual and area-based measures, contrasting the impact of income, occupation and education on survival.

## METHODS

### Patient cancer cohort

We analysed records from the national cancer registry[26] individually linked to the ONS-LS.[27] The LS sample is a random sample clustered by date of birth.[25 28] Census data for cohort members are available from the 1971 census through to the 2011 census. The ONS-LS also links life events data, including cancer registrations and deaths of members. The analysis cohort for this study included LS members present at either or both 2001 and 2011 census, and diagnosed with a first primary malignant cancer diagnosis between 1 January 2008 and 30 April 2016 at ages 20–100 years old. We examined five common cancer types: breast (International Classification of Diseases Version 10 (ICD-10) code C50), prostate (C61), colorectal (C18-21), non-Hodgkin's lymphoma (NHL) (C82–86) and bladder (C67). These specific malignancies were selected as it has been demonstrated that they exhibit significant area-based socioeconomic differentials for both sexes.[5] A small number (<20) of sex-site errors, and also a small number (<30) of men with breast cancer were excluded.

### Area-level deprivation

The Indices of Multiple Deprivation for England[29] and Wales[30] were used to measure area-based deprivation. We used the income, employment (ie, occupation) and education domains for the Lower-Layer Super Output Area (LSOAs, areas with a mean population of c.1500) of residence, using the temporally closest score to each census. For the 2001 census, this was the English Indices of Multiple Deprivation (IMD) 2004 and the Welsh metrics reported in 2005. For the 2011 census, this was the English IMD 2015 and the Welsh metrics reported in 2014. Each index was linked to the data as quintiles of the national distribution of areas, and three deprivation groups were created for the purposes of this analysis: least deprived (quintiles 1 and 2), mid (quintile 3) and most deprived (quintiles 4 and 5).

### Individual-level socioeconomic variables

Individual data on age, sex, qualifications and occupation were extracted directly from census data for each patient. Occupation type was derived using the three-group version of the National Statistics Socio-Economic Classification (NS-SEC) to ensure sufficient numbers to enable statistical analysis. These are 'technical, routine and manual occupations'; 'intermediate occupations';

or 'higher managerial, administrative and professional occupations'.[29]

Education level was categorised as one of three groups based on standard levels of English and Welsh qualifications used in the census: 'no qualifications'; 'school or college qualifications (General Certificates of Secondary Education (GCSEs), Advanced Levels (A-levels), apprenticeships, vocational qualifications or equivalent)'; or 'degree qualifications (degree-level education or higher)'.

Individual weekly income (GBP) was estimated indirectly from census data on an individual's age, sex and Standard Occupational Classification (SOC) code using an externally-validated linear model prediction method described by Clemens and Dibben.[31] We took a data-driven approach to adjust income for those aged over 60, who were most likely to be retired. We adjusted these income estimates using the observed annualised percentage decreases in income for those aged over 60 reported by the English Longitudinal Study of Ageing.[32] After applying this correction, income estimates were grouped into quintiles separately for each sex. LS members were then categorised into three groups by estimated income: lowest income (quintiles 1 and 2), middle income (quintile 3) and highest income (quintiles 4 and 5). Quintiles were calculated based on all available LS members (ie, not just patients with cancer), separately for each sex. Income estimates were therefore linked to occupation, however, the use of SOC codes rather than NS-SEC (as for the occupation variable above) means that these variables are independent of one another, since SOC codes are linked to specific jobs, as opposed to the broad NS-SEC categories for types of occupation.

Data were not available from the 2011 census for a small proportion of individuals; mostly accounted for by those who were diagnosed with cancer between 2008 and 2011 and died prior to the 2011 census.[25] Where possible, data from the 2001 census were used for these individuals. Missing data on qualifications or occupation (which includes long-term unemployed and students for the three-group version of the NS-SEC as recommended by the NS-SEC guidelines,[29] were completed where possible by proxy, using another adult resident in the household (usually household head). Following this procedure, 6%, <1% and 5% of records were missing individual deprivation data for occupation, education and income, respectively. These individuals were excluded.

## Survival analysis

Analyses were carried out separately for men and women. We analysed survival time (days between date of diagnosis and date of death or censoring) as a function of patient age and either socioeconomic group or area-based deprivation group, adjusted for the 'expected' mortality. Data were censored on the 31 December 2017, the date of the most recent linkage of the ONS-LS to mortality records.

We report net survival and 95% CIs, calculated using the non-parametric Pohar-Perme estimator[33] with the 'relsurv'[34] package in R V.3.6.3.[35] This is the most widely used, consistent estimator of net survival. Net survival is the survival probability patients would experience if their only possible cause of death were cancer. Net survival estimates are independent of underlying other-cause mortality and thus reflect cancer-specific prognosis. We account for underlying deaths from other causes using 'expected' mortality estimates for each individual socioeconomic group, which we extracted from life tables that we derived from this same ONS-LS cohort.[36] Expected mortality for the area-level deprivation analyses used life tables based on the overall ONS-LS cohort.

Net survival is reported as age-standardised estimates (Age-Standardised Net Survival, ASNS), derived using International Cancer Survival Standard (ICSS) weights[37] for age groups, with the youngest two groups merged together (ie, 15–54, 55–64, 65–74, 75+) to allow for the lower numbers in the youngest age groups in this population sample. For each deprivation or socioeconomic status measure and each cancer type, we calculated the arithmetic difference in survival between the most affluent and most deprived groups as the 'survival gap' (irrespective of which group displayed the highest or lowest survival).

Clustering of patients within geographical areas (LSOAs) was possible within the data, implying the need to take this into account in the analysis. However, an initial review of the data showed that a single-level analysis approach was sufficient here, as almost all individuals were unique to their LSOA within each cancer site. This applied to 96% of the analysis cohort; with the remaining 4% at a maximum of two individuals in the same geographical area. We therefore adopted a single-level approach.

### Patient and public involvement

This study was first presented to patient representatives at the National Cancer Resarch Institute (NCRI) Consumer Forum 'Dragon's Den' in 2017, where Mr Matthew Baker, along with nine other members of the public, provided input and ideas for the approach and methodology. Mr Baker has acted as non-academic co-investigator, helped to develop the study protocol and implement the research plans. He has attended all project meetings to provide insights on decision-making as the project progressed. Following the production of results the whole project team have worked on the dissemination and discussion of the results at both a further Dragon's Den meeting, and in online forums with policymakers. The research will also been presented online to members of the public who engaged with the topic via a specifically-targeted Facebook marketing campaign.

## RESULTS

Overall, 5551 men and 5284 women were included in the analyses. The cohort was broadly representative of the population from which it was drawn: the sex-specific

**Table 1** Distribution of patients with cancer in analysis cohort (N and %) compared with distribution (%) in England and Wales (E&W), patients diagnosed 1 January 2008 and 30 April 2016, by age group, cancer site and sex

| Cancer | Men | | | Women | | |
|---|---|---|---|---|---|---|
| | ONS-LS N | ONS-LS % | E+W* % | ONS-LS N | ONS-LS % | E+W* % |
| Breast | | | | | | |
| 20–54 | | | | 1050 | 30.3 | 32.0 |
| 55–64 | | | | 812 | 23.3 | 23.1 |
| 65–74 | | | | 834 | 24.0 | 21.2 |
| 75+ | | | | 777 | 22.4 | 23.7 |
| Total | | | | 3473 | 100.0 | 100.0 |
| Prostate | | | | | | |
| 20–54 | 109 | 3.6 | 3.9 | | | |
| 55–64 | 624 | 20.5 | 21.0 | | | |
| 65–74 | 1240 | 40.6 | 39.0 | | | |
| 75+ | 1071 | 35.2 | 36.1 | | | |
| Total | 3044 | 100.0 | 100.0 | | | |
| Colorectal | | | | | | |
| 20–54 | 142 | 9.3 | 9.4 | 130 | 10.5 | 11.0 |
| 55–64 | 325 | 21.4 | 20.1 | 233 | 18.8 | 17.0 |
| 65–74 | 486 | 31.9 | 31.8 | 343 | 27.7 | 25.9 |
| 75+ | 569 | 37.4 | 38.7 | 531 | 43.0 | 46.1 |
| Total | 1522 | 100.0 | 100.0 | 1237 | 100.0 | 100.0 |
| NHL | | | | | | |
| 20–54 | 90 | 18.6 | 19.7 | 71 | 19.0 | 16.7 |
| 55–64 | 108 | 22.4 | 19.8 | 62 | 16.7 | 18.7 |
| 65–74 | 141 | 29.2 | 27.9 | 110 | 29.6 | 27.3 |
| 75+ | 144 | 29.8 | 32.6 | 129 | 34.7 | 37.3 |
| Total | 483 | 100.0 | 100.0 | 372 | 100.0 | 100.0 |
| Bladder | | | | | | |
| 20–54 | 32 | 6.4 | 4.9 | 15 | 7.4 | 5.6 |
| 55–64 | 80 | 15.9 | 13.6 | 19 | 9.4 | 11.4 |
| 65–74 | 160 | 31.9 | 30.1 | 61 | 30.2 | 24.1 |
| 75+ | 230 | 45.8 | 51.4 | 107 | 53.0 | 58.9 |
| Total | 502 | 100.0 | 100.0 | 202 | 100.0 | 100.0 |

*Data Sources: National Cancer Registry Data, ONS-LS.
NHL, non-Hodgkin's lymphoma; ONS-LS, Office for National Statistics Longitudinal Study .

age distribution of cases for each cancer site were similar to that of the overall population of England and Wales (table 1). The data included a sufficiently large number of deaths by cancer and sex to enable net survival estimation (table 2). Similar proportions of men and women died within 1 and 5 years of diagnosis in the ONS-LS and in England and Wales.

Socioeconomic variations in net survival were observed at both 1 and 5 years after diagnosis for both sexes and for each cancer site. These results are displayed in figures 1–3. Survival tended to be negatively associated with area-level deprivation irrespective of the type

analysed, with estimates in the most deprived areas between 0.5% and 12.9% lower than in the least deprived areas at 1 year since diagnosis, and between 1.9% and 17.7% lower at 5 years. The only exceptions were for women with NHL, where area-based survival was not associated with increasing deprivation, and for men with colorectal cancer across occupation at 1 year. Differences across area-based income measures tended to show the most consistent and strongest negative associations. Patterns according to individual socioeconomic group were more mixed. The association between survival and deprivation was generally weaker for occupation than for

**Table 2** Number and percentage of men and women with each cancer type who died within 1 and 5 years of their diagnosis compared with England and Wales (E&W), patients diagnosed 1 January 2008 and 30 April 2016

| | Men | | | | | Women | | | | |
|---|---|---|---|---|---|---|---|---|---|---|
| Cancer | ONS-LS N | ONS-LS % 1 year | E+W* % 1 year | ONS-LS % 5 year | E+W* % 5 year | ONS-LS N | ONS-LS % 1 year | E+W* % 1 year | ONS-LS % 5 year | E+W* % 5 year |
| Breast | – | – | – | – | – | 3473 | 5 | 6 | 18 | 21 |
| Prostate | 3044 | 7 | 8 | 23 | 27 | – | – | – | – | – |
| Colorectal | 1522 | 23 | 25 | 47 | 51 | 1237 | 24 | 28 | 46 | 45 |
| NHL | 483 | 23 | 25 | 39 | 43 | 372 | 20 | 22 | 36 | 40 |
| Bladder | 502 | 25 | 28 | 51 | 55 | 202 | 39 | 41 | 56 | 59 |

*Data Sources: National Cancer Registry Data, ONS-LS.
NHL, non-Hodgkin's lymphoma; ONS-LS, Office for National Statistics Longitudinal Study .

other types of socioeconomic variables among men and, to a lesser extent, women. Percentage point differences in ASNS 1 year after diagnosis between individuals with degree-level qualifications and no qualifications ranged from 2.3% to 15.9%, among those with the highest and lowest incomes from −2.5% to 17.2%, and from −0.1% to 12.5% between those working in manual compared with professional occupations.

Differentials between individual-level socioeconomic groups in comparison to area-based deprivation quintiles are plotted against one another as the 'survival gap' in figure 4. The diagonal line indicates an equal extent of survival inequality measured in individual-level and area-level analysis. For men with colorectal and prostate cancer, the deprivation 'gap' was of a similar or slightly smaller magnitude between individual socioeconomic groups compared with area-based quintiles, for both 1-year and 5-year survival, for education, occupation and, to a lesser extent, income. Colorectal cancer differentials among women were greater using individual-based measures than area-based measures 1 year after diagnosis, but more similar 5 years after diagnosis, for all three types of deprivation. Breast cancer inequalities were of a similar magnitude 1 year after diagnosis for all types of deprivation, but larger for area-based measures after 5 years in comparison to those observed between individual socioeconomic groups. The deprivation gap tended to be smallest overall for men with prostate cancer and, to a lesser extent, women with breast cancer.

Bladder and NHL are lower incidence malignancies so the number of cases and deaths we examined were much smaller (tables 1 and 2). As such, the survival estimates for these cancers had wider CIs and the interpretation of these data should be treated with caution. Among men patterns for NHL were similar to the more common cancer sites. Among women with NHL an unexpected reverse trend was seen between area-based educational deprivation and survival, where more deprived women had better outcomes. More deprived bladder patients with cancer displayed poorer outcomes among both men and women. There was a suggestion that area-based measures had a greater impact compared with individual

socioeconomic group for men with bladder cancer, but patterns for women were similar between area-based and individual measures.

## DISCUSSION

We have described the differences in non-parametric univariable net survival for five cancers previously shown to have substantial area-level deprivation gaps in survival,[5] comparing inequalities derived using area-based deprivation measures to those obtained using individual measures of socioeconomic status. Consistent with the literature, survival was most often lower among those from more deprived localities irrespective of the type of deprivation analysed.[6 17] By contrast, there was an unexpected lack of overall trend of lower survival across the spectrum of individual socioeconomic groups as well as a notable lack of trend between individual income groups. Our results thus suggest that the role of individual characteristics ('person') versus area-based characteristics ('place') differs with respect to cancer outcomes and that the underlying reasons for this warrant further investigation.

### Individual versus area-based differentials

We calculated deprivation gaps in cancer survival in order to evaluate whether differentials between deprived and affluent individuals were larger, smaller or similar to those between deprived and affluent populations. The similarity of the magnitude of the deprivation gaps across area-based and individual-based measures suggests no evidence for a dilution effect, which was unexpected. Rather, these data are more supportive of the existence of two separate effects for cancer outcomes, one of 'person' (individual effect), another of 'place' (area-based effect). Our results are consistent with our previous findings which showed that deprived persons frequently resided in non-deprived areas,[25] and speaks against interpretations of area-based data where poorer health outcomes among deprived populations have been assumed to arise simply from poorer outcomes among deprived persons (dilution effect). The exception to this pattern is women's 1-year survival, where there is some suggestion of dilution for

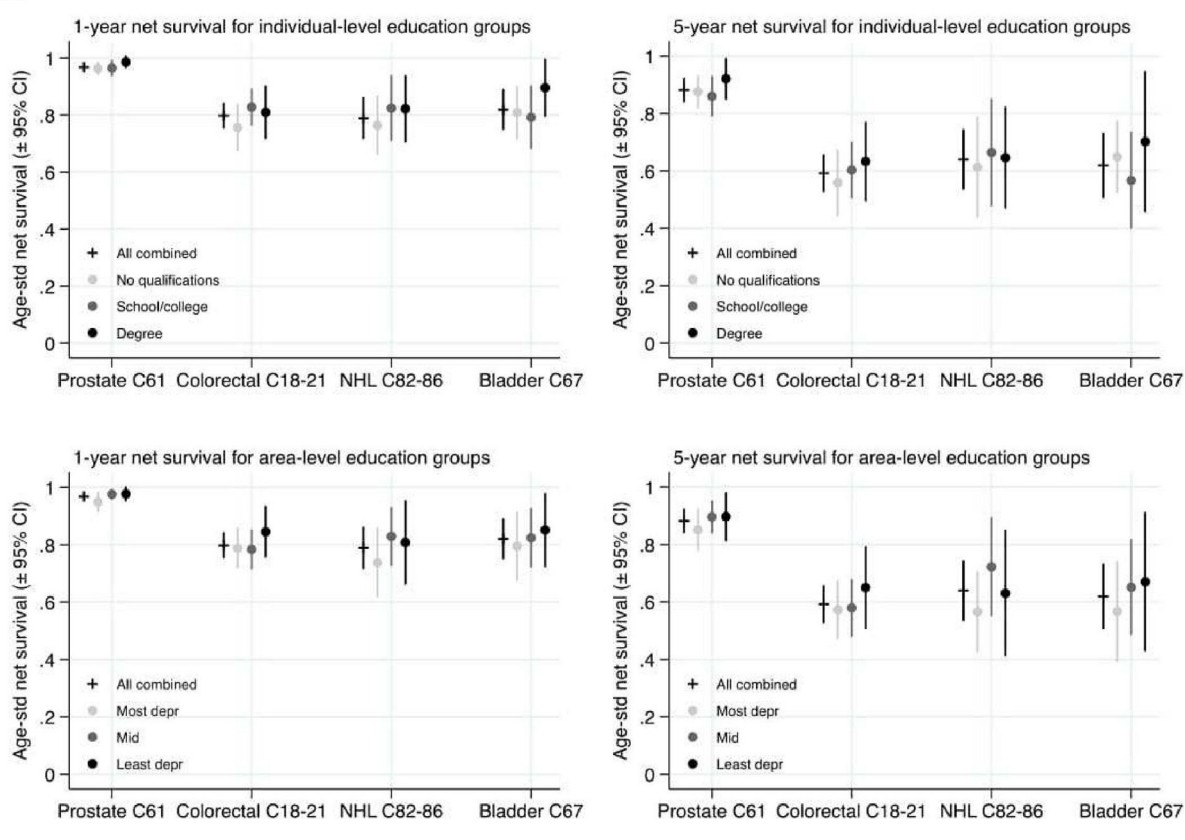

**A** Men

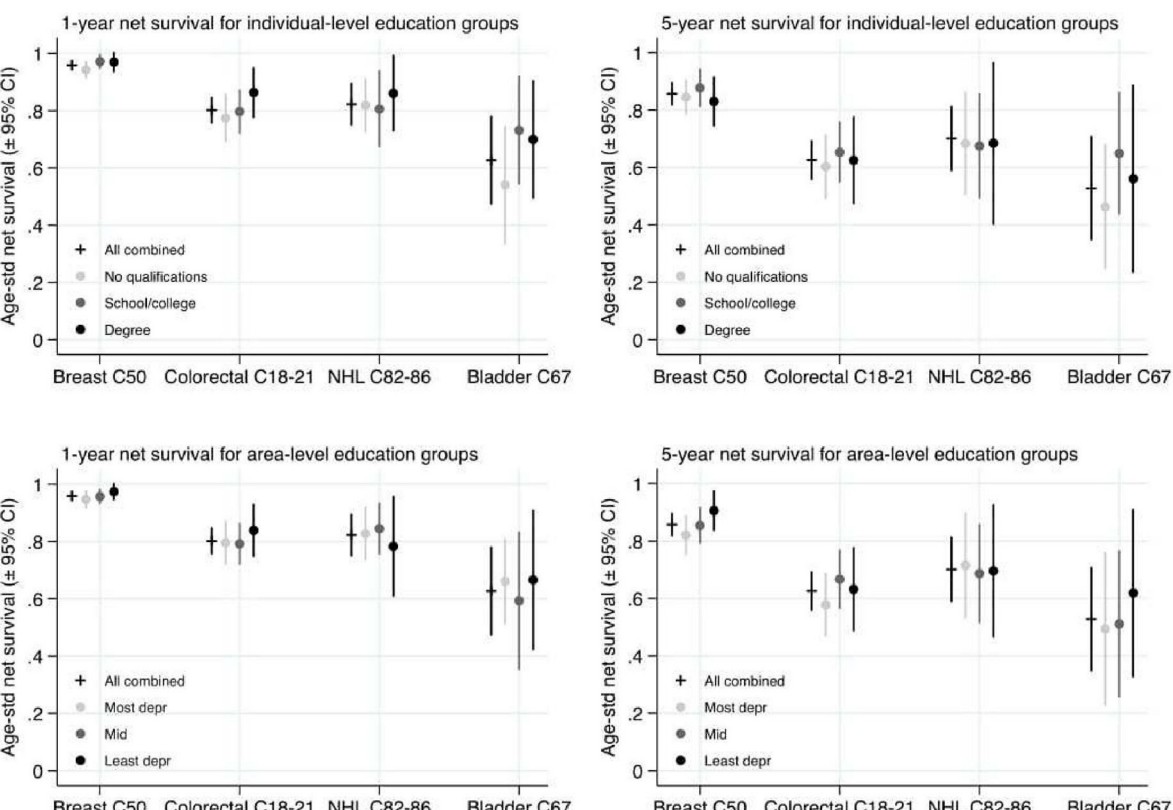

**B** Women

**Figure 1** Age-Standardised Net Survival estimates (95% CI) for individual-level compared to area-level measures of education: patients diagnosed 2008–2016. (A) Men and (B) Women.
Data source: Office for National Statistics Longitudinal Study. NHL, non-Hodgkin's lymphoma.

**A** Men

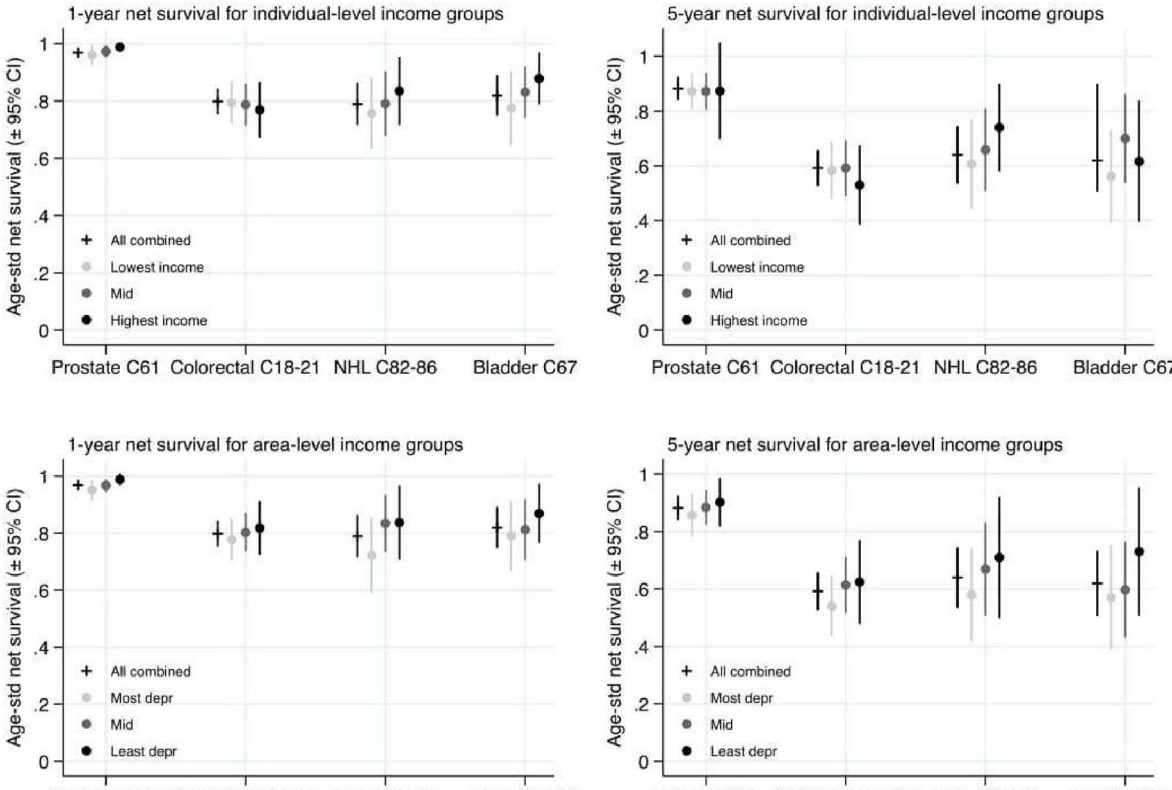

**B** Women

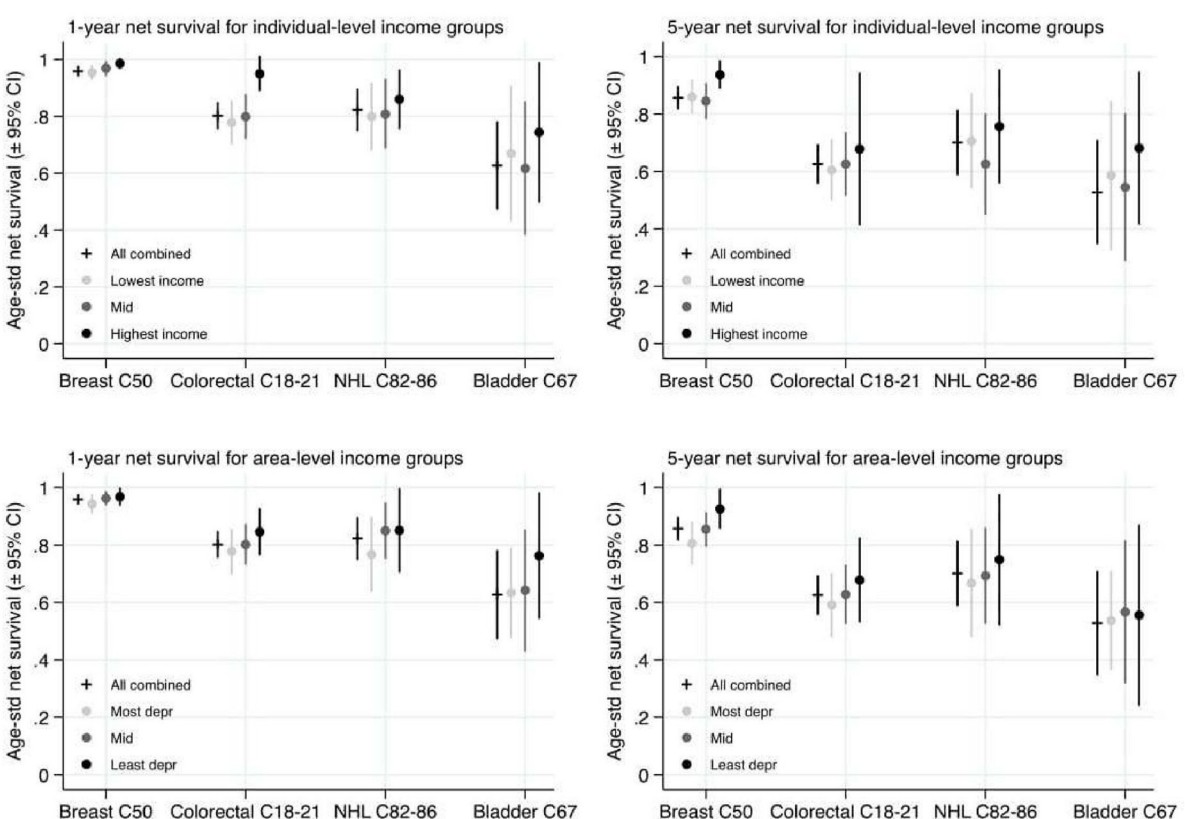

**Figure 2** Age-Standardised Net Survival estimates (95% CI) for individual-level compared to area-level measures of income: patients diagnosed 2008–2016. (A) Men and (B) Women.
Data source: Office for National Statistics Longitudinal Study. NHL, non-Hodgkin's lymphoma.

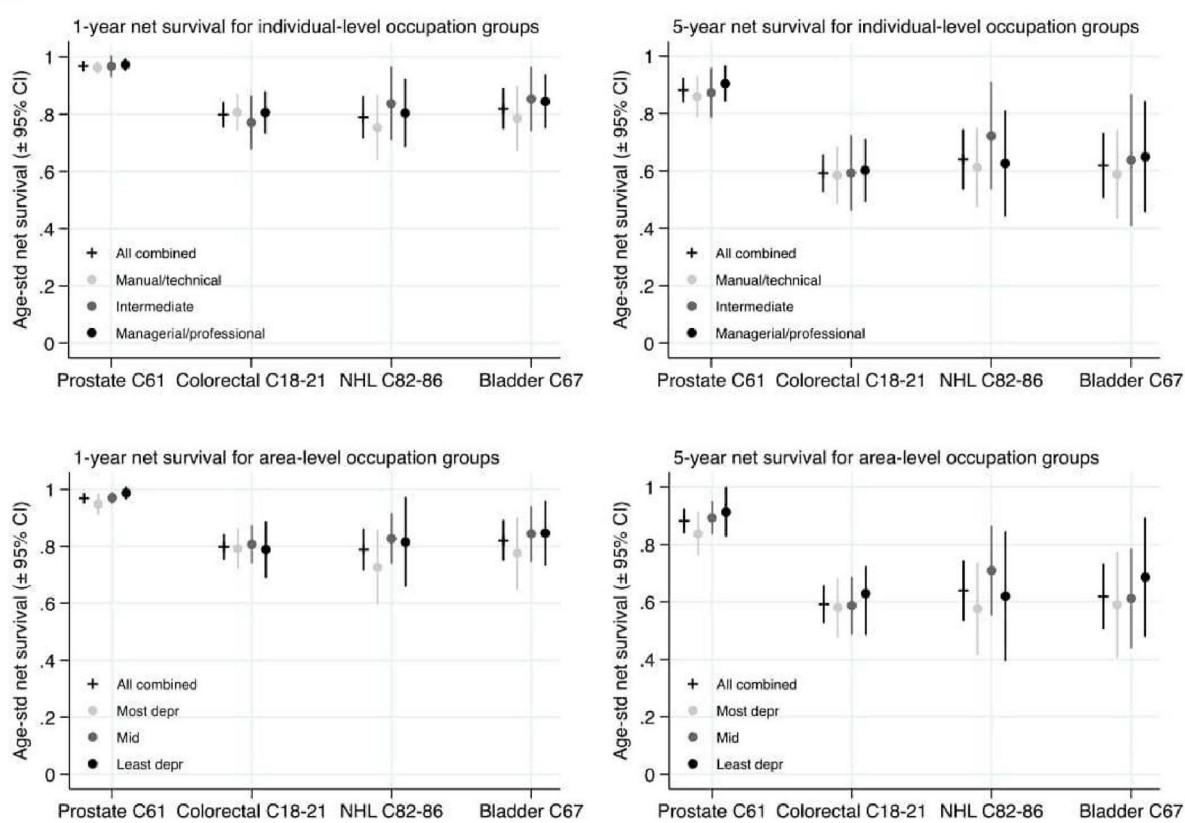

A   Men

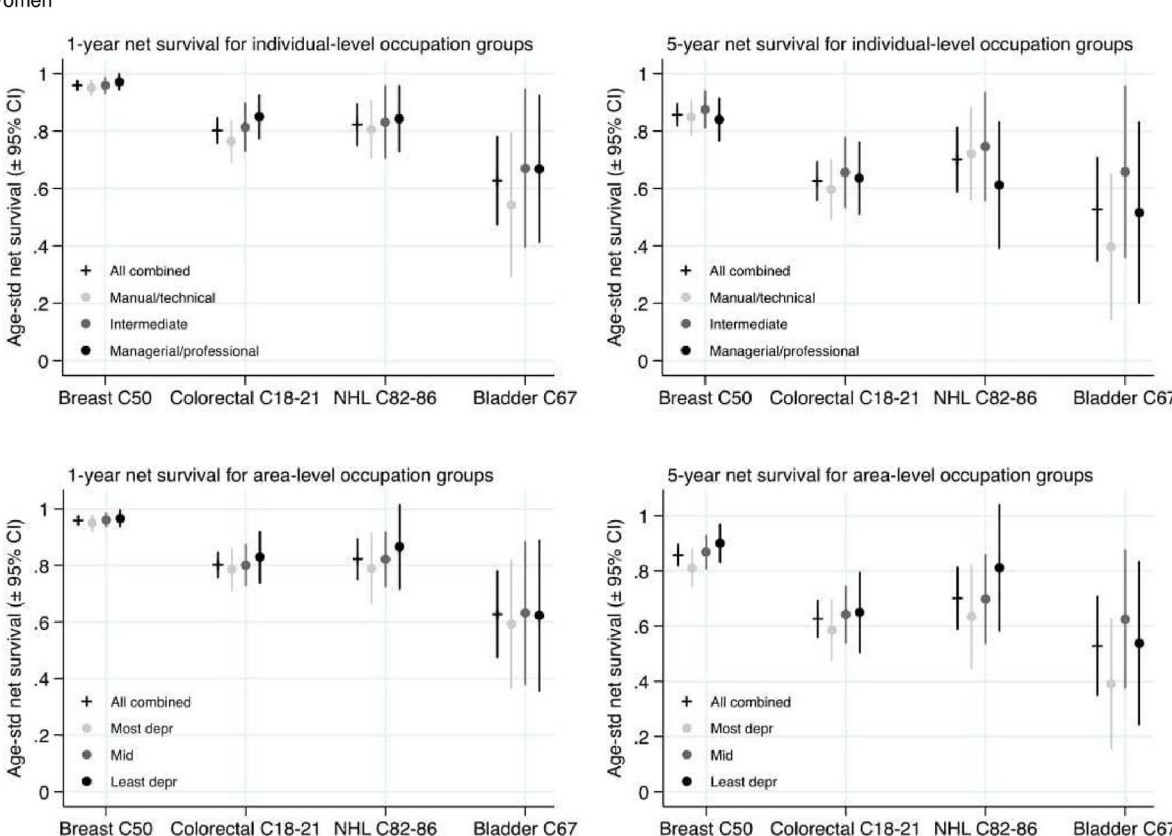

B   Women

**Figure 3**   Age-Standardised Net Survival estimates (95% CI) for individual-level compared with area-level measures of occupation: patients diagnosed 2008–2016. (A) Men (B) Women.
Data source: Office for National Statistics Longitudinal Study. NHL, non-Hodgkin's lymphoma.

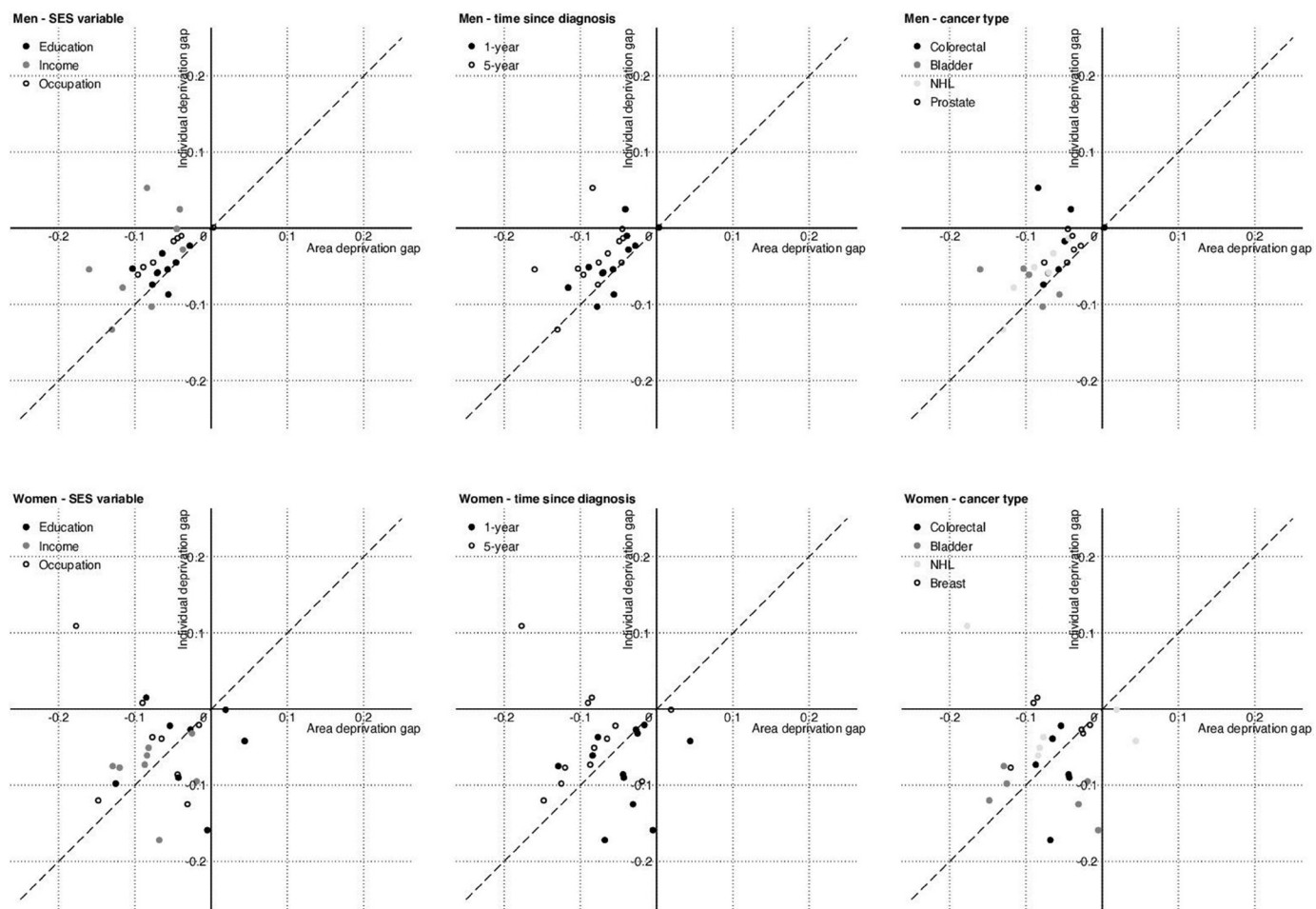

**Figure 4** Comparison of individual versus area-level deprivation gaps. *Deprivation gaps are negative where survival is lower in the more deprived groups. The dashed line indicates where the gap according to individual socioeconomic group and the area-level index is equal (ie, y=x).*
Data source: Office for National Statistics Longitudinal Study. NHL, non-Hodgkin's lymphoma; SES, socioeconomic status.

bladder and colorectal cancer, although these data points had wide CIs. The smaller differentials we observed for prostate and breast cancers are likely in part to constitute a form of ceiling effect, since differentials tend to be smaller when survival is high, even if the excess HR is of a similar magnitude to other cancers with lower survival.

### Domains of deprivation

For the most part the different measures of deprivation (income, education, occupation) exerted a broadly similar effect on cancer outcomes in area-based analyses. This has been previously observed[17] and is also consistent with sociological theory which states that socioeconomic status arises from three inter-related domains: a person's social class (broadly reflected by attained educational level), their social status or marker of prestige (broadly seen in the variety of occupations) and power (higher incomes affording a greater ability to spend and thus greater influence).[24] At the individual level, cancer outcomes were correlated with individual occupation and education for both men and women. Higher individual income among men was counter-intuitively associated with poorer outcomes in some analyses, and in others

displayed no discernible trend. These results were somewhat unexpected, especially given the clear association observed between area-based income deprivation and cancer outcomes in these same patients. These observations could be explained in part by the imputation of individual income from occupational codes. Alternatively, it is possible that this is a threshold effect, where variations in income above a certain level are not strongly associated with cancer outcomes. This suggests that the existing literature on income deprivation patterns may be picking out differentials between populations with differing proportions of persons on very low incomes.

### Strengths and limitations

Our study has a number of strengths. First, we used a unique, representative cohort of England and Wales within which we were able to classify individuals by both area-based deprivation score and individual socioeconomic group, so as to assess area-level and individual-level patterns within the same cohort of patients. We used newly derived individual life tables[36] to estimate underlying mortality for each individual socioeconomic group, matched to the most up-to-date methodology

for estimating non-parametric survival from cancer.[33] For prostate, breast and colorectal cancers we had data with sample sizes sufficiently large to generate relatively narrow CIs around our point estimates and so reasonably compare outcomes and the 'survival gaps' between the two different approaches. Numbers for bladder cancer and NHL were smaller (smallest group 202 cases with approximately 80 deaths) and the CIs much wider. These results should be interpreted with more caution: patterns for bladder cancer lent weight to our overall conclusions, while those for NHL were less consistent. Limitations to our approach include the use of a generic life table for the area-based analyses, as well as the need to estimate individual income on the basis of recorded occupation. Although life tables for England and Wales as a whole derived from quintiles of area-based income deprivation are available,[38] these have not been derived specifically for education and occupation subdomains. Further, our methods prioritised using life tables derived from the same cohort, so as not to introduce a bias from the use of a national life table: while the LS is representative of the overall population it is still only a small sample of the whole of England and Wales combined, and so it was more appropriate to extract observed rates of death from the cohort itself. The income variable for individuals was necessarily an estimate, since this information is not directly collected in the UK census. However, we used an externally-validated method,[31] which was based on a separate measure of occupation[29] to the employment domain, as well as age and sex, in order to generate the most accurate estimate as possible.

### Comparisons with published literature

Although there are now a substantially greater number of analyses which examine the impact of individual measures of socioeconomic status on cancer survival,[39] particularly from the Nordic countries, only two recent studies have described the impact of individual versus area-based measures on cancer survival as we do. In these studies, socioeconomic differentials according to individual and neighbourhood measures among men diagnosed with prostate cancer[40] and women diagnosed with breast cancer,[41] in the USA are similar to each other, supporting our own conclusions for these patients diagnosed in the UK.

### Policy implications and further research

Our results have significance for public health policy on inequalities, demonstrating that there is unlikely to be a simple correspondence between reducing differentials between more and less-deprived areas and improving outcomes for individually-deprived persons. While our analyses of these same data demonstrate evidence for contextual effects in some groups,[42] further research is required to establish the mechanisms by which these patterns arise. In particular, while area-based measures are exactly that, based on areas rather than individuals, they are derived from observed proportions of individual

people experiencing or having specific personal characteristics of low socioeconomic status within those areas (eg, the proportion of the population on income benefits, who are unemployed or lack of formal qualifications). As such, they measure both the influence of individual characteristics and reflect something of those individual's context, while they do not include any environmental measures of deprivation such as access to services, travel time, travel costs, number of GPs or specialist oncologists per capita. Nor do they take into account the way that healthcare is organised and delivered in different settings which is likely to be influential.[43] A further consideration is the social or community setting: our results do not and cannot measure factors such as social capital or social cohesion which may also be influential on an individual's ability to access the healthcare available to them. More detailed analyses are therefore required to better understand area-based patterns: first to further understand the nature and importance of 'place' in comparison to the influence of the socioeconomic status of the 'person', specifically as determined by the social, geographical and healthcare characteristics of areas in contrast to the characteristics of the individuals residing in them, and second to investigate the particular intrinsic characteristics of areas associated with poorer outcomes.

### CONCLUSION

We have conducted a unique analysis of cancer survival with respect to individual and area-based measures of deprivation. These data suggest that the influence of 'person' and 'place' on cancer outcomes warrants further investigation as part of a public health strategy to reduce cancer, as well as wider health, inequalities. These data support our other analyses on the differential effect of area, over and above the individual, as well as the particular characteristics of areas with poor outcomes would enable the derivation of more accurate hypotheses about the underlying causes of inequalities, and elucidate potential avenues for policy intervention.

**Author affiliations**
[1]Department of Non-Communicable Disease Epidemiology, London School of Hygiene & Tropical Medicine, London, UK
[2]School of Health and Social Care, Edinburgh Napier University, Edinburgh, UK
[3]National Cancer Registration and Analysis Service, Public Health England, London, UK
[4]National Cancer Research Institute Consumer Forum, London, UK

**Correction notice** This article has been corrected since it first published. Author name 'Lucy Elliss-Brookes' has been updated.

**Acknowledgements** The permission of the Office for National Statistics to use the Longitudinal Study is gratefully acknowledged, as is the help provided by staff of the Centre for Longitudinal Study Information & User Support (CeLSIUS). CeLSIUS is supported by the ESRC Census of Population Programme under project ES/V003488/1. The authors alone are responsible for the interpretation of the data in this paper. This work uses data provided by patients and collected by the National Health Service (NHS) as part of their care and support. Using patient data is vital to improve health and care for everyone. There is huge potential to make better use of information from people's patient records, to understand more about disease,

develop new treatments, monitor safety and plan NHS services. Patient data should be kept safe and secure, to protect everyone's privacy and it is important that there are safeguards to make sure that it is stored and used responsibly. Everyone should be able to find out about how patient data is used.

**Contributors** LMW, AB and IA conceived the study, developed the proposal, acquired the funding, ethical approval and data. They co-managed the conduct of the research and interpretation of the results. FCI conducted all analyses, assisted with the interpretation of the data and drafted the initial manuscript. LMW wrote the final version and led on the manuscript submission. LE-B and MB assisted in the acquisition of funding, the interpretation of the data and commented on drafts of the manuscript. All authors have approved the final version and agree to be accountable for all aspects of this work. LW is the guarantor who accepts full responsibility for the finished article and the conduct of the study. She had access to the data and controlled the decision to publish.

**Funding** This work was supported by the Economic and Social Research Council (ES/S001808/1).

**Competing interests** None declared.

**Patient and public involvement** Patients and/or the public were involved in the design, or conduct, or reporting, or dissemination plans of this research. Refer to the Methods section for further details.

**Patient consent for publication** Not applicable.

**Ethics approval** Ethics approval for this study was obtained from the London School of Hygiene & Tropical Medicine Ethics Online Application 14600; approved 01/02/2018. Data presented for England and Wales and presented in Tables 1 and 2 were obtained following statutory approval from the Confidentiality Advisory Group (CAG) of the Health Research Authority (HRA): PIAG 1–05(c) 2007. Informed consent was not required.

**Provenance and peer review** Not commissioned; externally peer reviewed.

**Data availability statement** Data may be obtained from a third party and are not publicly available. Data are not publicly available but can be accessed via appropriate application to the ONS Longitudinal Study (https://www.ucl.ac.uk/epidemiology-health-care/research/epidemiology-and-public-health/research/health-and-social-surveys-research-group/studies-44). This work contains statistical data from ONS which is Crown Copyright. The use of the ONS statistical data in this work does not imply the endorsement of the ONS in relation to the interpretation or analysis of the statistical data. This work uses research data sets which may not exactly reproduce National Statistics aggregates. Data presented for England and Wales in Tables 1 and 2 are available via application to the Public Health England Office for Data Release (https://www.gov.uk/government/publications/accessing-public-health-england-data/about-the-phe-odr-and-accessing-data).

**ORCID iDs**
Laura M Woods http://orcid.org/0000-0002-2178-1577
Aurélien Belot http://orcid.org/0000-0003-1410-5172
Iain M Atherton http://orcid.org/0000-0002-1822-3240
Lucy Elliss-Brookes http://orcid.org/0000-0003-1159-6607
Fiona C Ingleby http://orcid.org/0000-0003-1800-2015

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
