## [Reviewer comments · BMJ Open]

ARTICLE DETAILS

TITLE (PROVISIONAL)	Are deprivation-specific cancer survival patterns similar according to individual- and area-based measures? A cohort study of patients diagnosed with five malignancies in England & Wales, 2008-2016
AUTHORS	Woods, Laura; Belot, Aurelien; Atherton, Iain; Ellis-Brookes, Lucy; Baker, Matthew; Ingleby, Fiona

VERSION 1 – REVIEW

REVIEWER	Nelli Roininen University of Oulu, MRC Oulu
REVIEW RETURNED	16-Nov-2021

GENERAL COMMENTS	The research question is really important and worth studying. In Finland a recent study found bigger childhood cancer mortality in immigrants (Kyrölahti et al. 2018: Childhood cancer mortality and survival in immigrants: A population-based registry study in Finland). The problem of income-based health differences is big. The number of patients included in the study is large and the overall study design is good. Even the smallest cancer group included more than 200 patients and 80 deaths. You had to estimate the individual income of the study participants - the study would have been even more accurate with the actual incomes. Nevertheless, you discuss that and other limitations well and you don't make any conclusions not justified by results. The definition of abbreviation "SES" seems to be missing (page 9, row 60) - can you add that. Thank you about an interesting article and looking forward to other studies about this topic!
--

REVIEWER	Georgios Lyratzopoulos University College London, Department of Epidemiology & Public Health, Health Behaviour Research Centre
REVIEW RETURNED	31-Jan-2022

GENERAL COMMENTS	Thank you for asking me to review this study. I should declare that I am generally familiar with the work of the authors / and the ICON Group over the years, and know some of the well. I do not however perceive to have conflicts that preclude me from reviewing the study (I have no stakes in this genre of research anyhow other than as a reader), and I see my role more so like one of a (hopefully objective) 'internal examiner'. The study is really unique and one of a kind in the field – for the reasons summarised in the Introduction, which is excellently written. Not a comment that requires addressing, but I wanted to flag up a
---

classical paper on the matter of measurement of social class in epidemiology by Penny Liberatos et al. 1988
<https://academic.oup.com/epirev/article/10/1/87/553589?login=true>

I hope the research can be published as soon as possible. There are several areas that I think can be considered in way that will improve the manuscript and its resonance in the literature in the future.

High-level comments:

A. Possible confounding by health economy level variation in survival. The authors describe similarity in patterns of socioeconomic variation in incidence but without positing a strong interpretation framework – in my opinion. Some tentative suggestions are made regarding the influence of environmental/geographical variables (including social capital) – but I am not convinced these constitute strong hypotheses given the outcome examined (survival) – which is by its nature greatly influenced by both stage at diagnosis but also treatment access / quality / comprehensiveness. (Would have been a different story if the outcome was profiling of SES gradients in incidence, not survival, see for example <https://europepmc.org/article/med/35044985>). I wonder if confounding by health economy (e.g. at hospital / MDT / CCG level) may be at play, e.g. if ‘richer’ people (of whatever measure, direct or ecological) happen to be concentrated in health economies/systems with higher survival. See also this recent paper: Burton C, O'Neill L, Oliver P, Murchie P. Contribution of primary care organisation and specialist care provider to variation in GP referrals for suspected cancer: ecological analysis of national data. *BMJ Qual Saf.* 2020;29(4):296-303. <https://pubmed.ncbi.nlm.nih.gov/31586938/> - which although does not relate to survival but other metrics, reveals surprisingly strong ‘health economy’ effects above individual and general practice factors, and I think it is paradigmatically relevant. It is worth considering the following thought experiment: A country comprises just two ‘areas’ (north / south) with a hospital each. Imagine that all the rich people live in the north area, which therefore is also a ‘richer area’, and vice versa (poorer people live in the south, which also forms a poorer area). Now, let’s assume that survival in the north is better than the south, as the northern hospital manages patients more effectively than does the southern hospital. In that hypothetical scenario, I believe we will be able to observe highly similar patterns of SES variation in survival, no matter if we measure wealth at the individual level, or through ‘area’ measures. Could the authors generally consider the interpretation framework in view of likely role of differential concentration of different SES groups around hospitals / health economics /CCGs with different survival outcomes. I know ICON has done a lot of work over the years on variation in cancer survival by CCG. It would anyhow strengthen the paper if this hypothesis is considered even if to explain / posit that it is not likely, and explain why it is not. (Of course it may be a relevant hypothesis too). Similarly, I wonder if the authors should also examine variation in incidence, not only survival, as incidence of cancer is more likely to be ‘immune’ to confounding by health economy factors (treatment comprehensiveness) – obviously in a future, different, paper.

2. Inability to use multi-level modelling. The formal approach to examining the research question would have been the use of a 2-level model. I had to access another paper in this programme of work by co-authors
<https://bmcpublikealth.biomedcentral.com/track/pdf/10.1186/s12889->

022-12525-1.pdf in order to appreciate that using a 2-level model (patients nested within small areas) was not possible due to nearly always having only one patient per small area (in this sample). Please introduce this earlier on – indicating what the ‘ideal’ study would have been (through use of multi-level modelling) and then explain why this was not possible and the approach taken. This issue has implications for some of the language used to introduce / interpret the findings, and for the design of future studies.

3. I think the prior work of the authors on cross-correlations between the measures examined <https://bmjopen.bmj.com/content/bmjopen/10/11/e041714.full.pdf> and also the paper above need to be alluded to more clearly in the background. I would also encourage a little (few-line) summary of the two previous papers (see above) in the literature comparisons section – so that prior work is credited and the value of enrichment from the current work more clearly appreciated. I actually cannot find a section entitled ‘comparisons with literature’ in Discussion – I think it would be nice to add. Related to this, is there US or other country literature, by any chance, that is of relevance?

Some more specific comments.

1. Abstract-objective:

a) The study title is commendably exact and accurate given the methods used and the interpretation of the findings. In contrast, the Abstract, Objective reads: “To investigate the relative influence of individual- (person) vs. area-based (place) measures of deprivation” – I think the term ‘relative’ and ‘versus’ here sound to be too strong, given that it there were formal direct comparisons of the two components (person-level and area-level variables). E.g. such as would have been the case if it were possible to use multi-level modelling that would have helped to formally partition their influence and ‘relative’ importance of individual ‘vs’ small area measures (see also above re multi-level model). A more accurate I think expression for the Objective, given the methods that it was possible to employ, would be something like: “To investigate whether gradients in net survival are similar when either person-level or area-level measures of income, deprivation and occupation are used”. I.e. something similar to the title actually.

b) Similarly, I have important reservations about the use of the term ‘place’ please see below.

2. Abstract-conclusion: “Conclusion: Further research should address the existence of contextual effects using a modelling approach as well as define the particular characteristics of areas with poor outcomes in order to inform policy intervention.” Two points:

a) I was not sure what the authors mean by contextual effects. I see in their paper by Ingleby et al. BMC Cancer 2020 they define contextual effects as interactions between person- and area-level variables – and examined such effects, finding a rather mixed picture for their presence/absence. Is this about expanding to more cancers? Or about using multi-level modelling. These are all possible interpretations. Please disambiguate as current version is not totally clear.

b) Also the latter part of the argument, ‘to define characteristics of areas with poor outcomes’ to me does not obviously follow from the

findings and would require quite a similar kind of analysis (again multi-level modelling would be useful). Perhaps I am missing the point but please consider. [Noting that the hospital(s) covering an area is also one of its area 'characteristics' – see above, point A].

3. Page 7, lines 25-26, please explain a bit more about the method used to assign income to individuals using the Clemens and Dibben method (which was then further adjusted using the ELSA data in over 60s). Key components of the Clemens and Dibben method should be described in a line or two – important as this also comes up later on in the limitations.

4. Page 9 of the pdf, line 11, I think 'variable' needs to be in plural.

5. Page 10, line 43, should 'state' read 'status'? I am actually not familiar with the juxtapositions of class to education, status to occupation, and power to income. Do these 1:1 suggested relationships represent propositions by the authors themselves or they based on other consensus? I do not know this area well enough but at face value I am not sure that class does not have to do with all three of education/occupation/income, not only education – etc. similarly for the other juxtapositions.

6. Page 11, strengths and limitations.

a. The authors are entirely correct in their exposition of the uniqueness of the dataset and the use of a 'from within the cohort' life table to use in relative survival – these are great strengths.

b. For me the biggest limitation seems to be the fact that the sample size precluded use of multi-level models, due to very small number of LSOAs with more than one patient (as per Ingleby et al. BMC 2020). Can this be mentioned and theorised about clearly – this is important for future research priorities / design where samples are adequate, and also increases the general education and information value of the paper.

c. Still on sample size, the authors mention: "For prostate, breast and colorectal cancers we were able to obtain sample sizes sufficiently large to confidently compare outcomes and the 'survival gaps' between the two different approaches." Please expand on what is meant – was a test of significance / formal statistical assessment used to 'confidently compare' the survival gaps? Actually, this problematic is better in Methods? Or even Results? Please consider.

d. Also "Numbers for bladder cancer and non-Hodgkin lymphoma were smaller (smallest group 202 cases with approximately 80 deaths)." This sounds appropriate but could the authors posit even in broad terms, what sample size might have been adequate / needed?

7. Page 11, "Policy implications and further research". The authors are correct in reflecting that the area-based measures used are derived as aggregates of measure pertaining to individuals per se, and that they do not incorporate genuine geographical (e.g. environmental pollution/urban landscape/transport/recreation facilities/proximity to schools or gyms etc.) variables. In spite of this the paper often talks about 'place' (as in Conclusions). To most readers 'place' denotes the truly geographical properties of an area, or at least both local population + geography, so I would urge authors to reconsider the use of the term 'place' throughout – are they really

	referring to 'places'? For sure it would be good to clarify what is meant by 'place' early on in the paper, if the term survives the rewrite; alternatively perhaps use an alternative, as essentially we are talking about small area population clusters, not really spaces / places / environment. 8. Page 26 – I could not quite find the data / information relating to these figure exhibits presented in the main text of Results – apologies if I missed it but others will too as currently. Thank you for the great work which definitely raises many interesting questions that need to be addressed by further research – the paper provides very novel evidence of great importance to the likely mechanisms by which cancer survival inequalities are generated , and hope to be part of the published literature soon.
--	--

VERSION 1 – AUTHOR RESPONSE

Reviewer: 1 - Dr. Nelli Roininen, University of Oulu, Pohjois-Pohjanmaan Sairaanhoidopiiri

The research question is really important and worth studying. In Finland a recent study found bigger childhood cancer mortality in immigrants (Kyrönlahti et al. 2018: Childhood cancer mortality and survival in immigrants: A population-based registry study in Finland). The problem of income-based health differences is big.

The number of patients included in the study is large and the overall study design is good. Even the smallest cancer group included more than 200 patients and 80 deaths. You had to estimate the individual income of the study participants - the study would have been even more accurate with the actual incomes. Nevertheless, you discuss that and other limitations well and you don't make any conclusions not justified by results.

The definition of abbreviation "SES" seems to be missing (page 9, row 60) - can you add that.

Thank you about an interesting article and looking forward to other studies about this topic!

We thank Dr Roininen for these positive comments. Since this is the only occurrence of SES in the paper we have replaced with the full term (socioeconomic status) (*RESULTS: Survival analysis, paragraph 3*)

Reviewer: 2 - Dr. Georgios Lyratzopoulos, University College London

Thank you for asking me to review this study. I should declare that I am generally familiar with the work of the authors / and the ICON Group over the years, and know some of them well. I do not however perceive to have conflicts that preclude me from reviewing the study (I have no stakes in this genre of research anyhow other than as a reader), and I see my role more so like one of a (hopefully objective) 'internal examiner'.

The study is really unique and one of a kind in the field – for the reasons summarised in the Introduction, which is excellently written. Not a comment that requires addressing, but I wanted to flag up a classical paper on the matter of measurement of social class in epidemiology by Penny Liberatos et al. 1988 <https://academic.oup.com/epirev/article/10/1/87/553589?login=true>

Thank you for flagging up this paper, we note it for future reference.

I hope the research can be published as soon as possible. There are several areas that I think can be considered in way that will improve the manuscript and its resonance in the literature in the future.

High-level comments:

1. Possible confounding by health economy level variation in survival. The authors describe similarity in patterns of socioeconomic variation in incidence but without positing a strong interpretation framework – in my opinion. Some tentative suggestions are made regarding the influence of environmental/geographical variables (including social capital) – but I am not convinced these constitute strong hypotheses given the outcome examined (survival) – which is by its nature greatly influenced by both stage at diagnosis but also treatment access / quality / comprehensiveness. (Would have been a different story if the outcome was profiling of SES gradients in incidence, not survival, see for example <https://europepmc.org/article/med/35044985>). I wonder if confounding by health economy (e.g. at hospital / MDT / CCG level) may be at play, e.g. if ‘richer’ people (of whatever measure, direct or ecological) happen to be concentrated in health economies/systems with higher survival. See also this recent paper: Burton C, O’Neill L, Oliver P, Murchie P. Contribution of primary care organisation and specialist care provider to variation in GP referrals for suspected cancer: ecological analysis of national data. *BMJ Qual Saf.* 2020;29(4):296-303. <https://pubmed.ncbi.nlm.nih.gov/31586938/> - which although does not relate to survival but other metrics, reveals surprisingly strong ‘health economy’ effects above individual and general practice factors, and I think it is paradigmatically relevant. It is worth considering the following thought experiment: A country comprises just two ‘areas’ (north / south) with a hospital each. Imagine that all the rich people live in the north area, which therefore is also a ‘richer area’, and vice versa (poorer people live in the south, which also forms a poorer area). Now, let’s assume that survival in the north is better than the south, as the northern hospital manages patients more effectively than does the southern hospital. In that hypothetical scenario, I believe we will be able to observe highly similar patterns of SES variation in survival, no matter if we measure wealth at the individual level, or through ‘area’ measures. Could the authors generally consider the interpretation framework in view of likely role of differential concentration of different SES groups around hospitals / health economics / CCGs with different survival outcomes. I know ICON has done a lot of work over the years on variation in cancer survival by CCG. It would anyhow strengthen the paper if this hypothesis is considered even if to explain / posit that it is not likely, and explain why it is not. (Of course it may be a relevant hypothesis too). Similarly, I wonder if the authors should also examine variation in incidence, not only survival, as incidence of cancer is more likely to be ‘immune’ to confounding by health economy factors (treatment comprehensiveness) – obviously in a future, different, paper.

We thank you for these comments and the higher level thinking that has been undertaken! We may have misunderstood a little of what is intended here but we respond as follows;

- The focus of this paper is the reporting of non-parametric, univariable survival estimates, thus it is principally descriptive and not thus able to examine confounding per se. We have published a more extensive, multivariable analyses for three out of these six sites (for which numbers permitted these more complex analyses) which seeks to disentangle the compositional effect (individuals with different SES within the same context) from the contextual effect (the effect of area upon individuals).
- That said, we agree that there is likely to be an impact of place with respect to the healthcare available and the way it is utilised by both individuals and healthcare providers. The paper helpfully cited by the reviewer supports elements of our Discussion which propose why there might be an effect of ‘place’ over ‘person’. We have modified the Discussion to highlight this including the addition of this reference (*DISCUSSION, Policy implications & further research*).

- Whilst variations in cancer incidence would be interesting this is not the main interest here: incidence patterns are likely to be highly socio-economically varied with regards to individual health behaviours (diet, smoking etc.) as well as geographically varying risk factors (air pollution, food deserts etc.) as well as health economy factors (screening availability, GPs per head, referral speeds etc.).
2. Inability to use multi-level modelling. The formal approach to examining the research question would have been the use of a 2-level model. I had to access another paper in this programme of work by co-authors <https://bmcpublichealth.biomedcentral.com/track/pdf/10.1186/s12889-022-12525-1.pdf> in order to appreciate that using a 2-level model (patients nested within small areas) was not possible due to nearly always having only one patient per small area (in this sample). Please introduce this earlier on – indicating what the ‘ideal’ study would have been (through use of multi-level modelling) and then explain why this was not possible and the approach taken. This issue has implications for some of the language used to introduce / interpret the findings, and for the design of future studies.

Indeed we had always intended that the main analyses in this Project would be the modelling of these data using a multi-level approach and the software *mexhaz* (as reported in the BMC Public Health article). However, as you have spotted this was not possible, or, more importantly, necessary due to the final distribution of cases across small areas. By contrast, the focus of this current paper is the comparison of non-parametric, univariable estimates of net survival (not multivariable modelling of the excess hazard to look at the association between prognostic factors and excess hazard/net survival). Hence the issue of multi-level models is of less direct relevance. The treatment of the (potentially) clustered data is, however, still important. Accordingly we have added comments relating to this to the Methodology section (*METHODS: Survival analysis, final paragraph*).

3. I think the prior work of the authors on cross-correlations between the measures examined <https://bmjopen.bmj.com/content/bmjopen/10/11/e041714.full.pdf> and also the paper above need to be alluded to more clearly in the background. I would also encourage a little (few-line) summary of the two previous papers (see above) in the literature comparisons section – so that prior work is credited and the value of enrichment from the current work more clearly appreciated. I actually cannot find a section entitled ‘comparisons with literature’ in Discussion – I think it would be nice to add. Related to this, is there US or other country literature, by any chance, that is of relevance?

Thank you for highlighting this. Our paper on the concordance between individual and area-based measures of SES is the subject of the last paragraph of the background, but we have strengthened this by rephrasing this slightly. (*INTRODUCTION, final paragraph*). We have further cited our second analysis in the ‘*Policy implications & further research*’ section and have enhanced the Discussion with a section describing the limited amount of international literature in this area (*DISCUSSION: Policy implications & further research, Comparisons with published literature*).

Some more specific comments.

- 1) Abstract-objective:
 - a. The study title is commendably exact and accurate given the methods used and the interpretation of the findings. In contrast, the Abstract, Objective reads: “To investigate the relative influence of individual- (person) vs. area-based (place) measures of deprivation” – I think the term ‘relative’ and

'versus' here sound to be too strong, given that it there were formal direct comparisons of the two components (person-level and area-level variables). E.g. such as would have been the case if it were possible to use multi-level modelling that would have helped to formally partition their influence and 'relative' importance of individual 'vs' small area measures (see also above re multi-level model). A more accurate I think expression for the Objective, given the methods that it was possible to employ, would be something like: "To investigate whether gradients in net survival are similar when either person-level or area-level measures of income, deprivation and occupation are used". I.e. something similar to the title actually.

Thank you for this comment. We have modified to "To investigate if measured inequalities in cancer survival differ when using individual- ('person') compared to area- ('place') based measures of deprivation for three socio-economic dimensions: income, deprivation and occupation" (*ABSTRACT*).

- b. Similarly, I have important reservations about the use of the term 'place' please see below.
Please see responses below.

- 2) Abstract-conclusion: "Conclusion: Further research should address the existence of contextual effects using a modelling approach as well as define the particular characteristics of areas with poor outcomes in order to inform policy intervention."

Two points:

- a. I was not sure what the authors mean by contextual effects. I see in their paper by Ingleby et al. BMC Cancer 2020 they define contextual effects as interactions between person- and area-level variables – and examined such effects, finding a rather mixed picture for their presence/absence. Is this about expanding to more cancers? Or about using multi-level modelling. These are all possible interpretations. Please disambiguate as current version is not totally clear.

Our intended meaning was indeed the same as the Ingleby paper. We have modified this section to make this clearer (*ABSTRACT*).

- b. Also the latter part of the argument, 'to define characteristics of areas with poor outcomes' to me does not obviously follow from the findings and would require quite a similar kind of analysis (again multi-level modelling would be useful). Perhaps I am missing the point but please consider. [Noting that the hospital(s) covering an area is also one of its area 'characteristics' – see above, point A].

We have been more precise with the language here, specifying the types of characteristics we have in mind (*ABSTRACT*).

- 3) Page 7, lines 25-26, please explain a bit more about the method used to assign income to individuals using the Clemens and Dibben method (which was then further adjusted using the ELSA data in over 60s). Key components of the Clemens and Dibben method should be described in a line or two – important as this also comes up later on in the limitations.

This has now been expanded to provide further detail (*METHODS: Individual-level socio-economic variables, paragraph 3*).

- 4) Page 9 of the pdf, line 11, I think 'variable' needs to be in plural.

This has been corrected.

- 5) Page 10, line 43, should 'state' read 'status'? I am actually not familiar with the juxtapositions of class to education, status to occupation, and power to income. Do these 1:1 suggested relationships represent propositions by the authors themselves or they based on other consensus? I do not know this area well enough but at face value I am not sure that class does not have to do with all three of education/occupation/income, not only education – etc. similarly for the other juxtapositions.

Thanks you for this comment, the typo on 'status' has been corrected. The dimensions of SES derive originally from the work of sociologist Max Weber (referenced) although they have been variously interpreted and re-applied since in the sociological literature. We do not have space here to investigate these various debates, but we draw attention to the original history of these ideas in order to highlight the theoretical roots of three very commonly used measures. We have expanded the parentheses slightly with the aim of making this a little more comprehensible to an epidemiological audience (*DISCUSSION: Domains of deprivation*).

- 6) Page 11, strengths and limitations.
- a. The authors are entirely correct in their exposition of the uniqueness of the dataset and the use of a 'from within the cohort' life table to use in relative survival – these are great strengths.
Thank you for this note of confidence.
 - b. For me the biggest limitation seems to be the fact that the sample size precluded use of multi-level models, due to very small number of LSOAs with more than one patient (as per Ingleby et al. BMC 2020). Can this be mentioned and theorised about clearly – this is important for future research priorities / design where samples are adequate, and also increases the general education and information value of the paper.
As detailed above, our aim in this paper was not to directly model the contextual effect or the clustering effects, but simply to describe the differentials observed using the two types of SES measure. We have highlighted this in response to high-level comment number 2.
 - c. Still on sample size, the authors mention: "For prostate, breast and colorectal cancers we were able to obtain sample sizes sufficiently large to confidently compare outcomes and the 'survival gaps' between the two different approaches." Please expand on what is meant – was a test of significance / formal statistical assessment used to 'confidently compare' the survival gaps? Actually, this problematic is better in Methods? Or even Results? Please consider.
Please see next comment below.
 - d. Also "Numbers for bladder cancer and non-Hodgkin lymphoma were smaller (smallest group 202 cases with approximately 80 deaths)." This sounds appropriate but could the authors posit even in broad terms, what sample size might have been adequate / needed?
These two comments essentially come down to a subjective assessment of the CIs rather than a direct comparison with a formal sample size calculation, since such power calculations or hypothesis testing would not be appropriate in our descriptive analysis. We have clarified this in the text. (*DISCUSSION: Strengths and limitations*).

- 7) Page 11, “Policy implications and further research”. The authors are correct in reflecting that the area-based measures used are derived as aggregates of measure pertaining to individuals per se, and that they do not incorporate genuine geographical (e.g. environmental pollution/urban landscape/transport/recreation facilities/proximity to schools or gyms etc.) variables. In spite of this the paper often talks about ‘place’ (as in Conclusions). To most readers ‘place’ denotes the truly geographical properties of an area, or at least both local population + geography, so I would urge authors to reconsider the use of the term ‘place’ throughout – are they really referring to ‘places’? For sure it would be good to clarify what is meant by ‘place’ early on in the paper, if the term survives the rewrite; alternatively perhaps use an alternative, as essentially we are talking about small area population clusters, not really spaces / places / environment.
- We prefer to retain these terms as they have been used throughout the project and correspond to terminology already published with other outputs/dissemination activities. We have, however, clarified what we mean by each (place = the measured characteristics of the small area within which a person resides, person = the particular personal characteristics of the individual) (*ABSTRACT, DISCUSSION, paragraph 1*), and have ensured that the terms are consistently used and explained/explicit throughout.
- 8) Page 26 – I could not quite find the data / information relating to these figure exhibits presented in the main text of Results – apologies if I missed it but others will too as currently.

This is the subject of what is now the second paragraph of the results section. We have now highlighted the correspondence between this text and the figures more explicitly (*Results, paragraph 2*).

Thank you for the great work which definitely raises many interesting questions that need to be addressed by further research – the paper provides very novel evidence of great importance to the likely mechanisms by which cancer survival inequalities are generated , and hope to be part of the published literature soon.

VERSION 2 – REVIEW

REVIEWER	Georgios Lyratzopoulos University College London, Department of Epidemiology & Public Health, Health Behaviour Research Centre
REVIEW RETURNED	28-Apr-2022
GENERAL COMMENTS	Thank you to the authors for engaging with and addressing all comments. Particularly regarding healthcare factors now being listed as possible contributors to 'place' effects, alongside other likely influences. I look forward to seeing this important paper published.